# Power Loss Minimization and Voltage Profile Improvement by System Reconfiguration, DG Sizing, and Placement

**Mlungisi Ntombela \*, Kabeya Musasa and Moketjema Clarence Leoaneka**

Department of Electrical Power Engineering, Faculty of Engineering and the Built Environment, Durban University of Technology, Durban 4000, South Africa

\* Correspondence: 21210920@dut4life.ac.za

**Abstract:** A number of algorithms that aim to reduce power system losses and improve voltage profiles by optimizing distributed generator (DG) location and size have already been proposed, but they are still subject to several limitations. Hence, new algorithms can be developed or existing ones can be improved so that this important issue can be addressed more appropriately and effectively. This study proposes a reconfiguration methodology based on a hybrid optimization algorithm, consisting of a combination of the genetic algorithm (GA) and the improved particle swam optimization (IPSO) algorithm for minimizing active power loss and maintaining the voltage magnitude at about 1 p.u. The buses at which DGs should be injected were identified based on optimal real power loss and reactive power limit. When applying the proposed optimization algorithm for DGs allocation in power system, the search space or number of iterations was reduced, increasing its convergence rate. The proposed reconfiguration methodology was test in an IEEE-30 bus electrical network system with DGs allocations and the simulations were conducted using MATLAB software compared to other optimization algorithms, such as GA, PSO, and IPSO, the combination of GA and IPSO or Hybrid GA & IPSO (HGAIPSO) method has a smaller number of iterations and is more effective in optimization problems. The effectiveness of the proposed HGAIPSO has been tested on IEEE-30 bus network system with DGs allocations, and the obtained test results have been compared to those from other methods (i.e., GA, PSO, and IPSO). The simulation results show that the proposed HGAIPSO can be an efficient and promising optimization algorithm for distribution network reconfiguration problems. The IEEE-30 bus test system with DGs integrated at various location revealed reductions in overall real power loss of 40.7040%, 36.2403%, and 42.9406% for type 1, type 2, and type 3 DGs allocation, respectively. The highest bus voltage profile goes to 1.01 pu in the IEEE-30 bus.

**Keywords:** power loss minimization; power system reconfiguration; voltage profile improvement; optimization algorithms; reduction of power loss; distribution power network optimization

## 1. Introduction

It has always been necessary for power distribution networks to adapt to variations in load demand, which has led to voltage oscillations beyond the permissible variation range at various buses and power losses. As a result, proper placement and scale of distributed generation (DG) are required to improve the voltage profile and reduce electrical power losses. According to research, global consumption is predicted to expand at a 1.6 percent yearly rate between now and 2025. Consequently, distributed generation (DG), also known as alternative energy systems, is likely to play a larger role in the future of power systems [1]. Because of their overall favorable impacts on power networks, DG units are now becoming more in use in electrical distribution networks. For smart grid technology, DG systems constitute the backbone of smart electrical networks. The DG systems can also increase system dependability by acting as a backup generator for some customers in the event of electricity outages [2].

There are two categories of DG technologies: those that use fossil fuels and those that use renewable energy sources. Fossil-fuel-based DGs include internal combustion engines,

combustion turbines, and fuel cells. Examples of renewable energy sources include modest hydro, solar, biomass, geothermal, wind turbines, and various DGs based on these sources. The technological effects of DG on electrical networks must be evaluated. It takes time to evaluate the technical effects of DG on power networks [3]. To prevent power losses and changes in voltage profile, DGs placed in electrical distribution networks must be linked carefully. Factors such as fault currents, voltage oscillations, interference in voltage control procedures, increased power losses, increased system capital, and operational expenses, etc. may result from a weak DG location and allocated capacity. The installation of DG units in power systems is not a straightforward decision; therefore, the placement and sizing of DG units to reduce losses and to improve voltage profile must be carefully considered. Different optimization methods for DG allocations in electrical distribution network are being developed for power loss minimization and voltage profile improvement.

Changes in load demand have long been a problem for the power distribution network, leading to voltage oscillations outside the range allowed by fluctuations at various buses and power losses. To enhance the voltage profile and lower electrical power losses, distributed generation (DG) must be placed and scaled properly. Research indicates that between now and 2025, global consumption will grow at a 1.6 percent annual rate. Consequently, distributed generation (DG), usually referred to as alternative energy systems, will likely have a bigger impact on power networks in the future [4]. DG units are being utilized more commonly in electrical distribution networks because of their overall positive effects on power networks. DG systems serve as the foundation of smart electrical networks in the context of smart grid technologies. In the case of power outages, these DG systems can serve as backup generators for select consumers, which helps boost system reliability.

### 1.1. Context, Background, and Motivation

Over 6000 MW of generation capacity has been added to the existing power systems in African countries via renewable energy sources, namely the wind, solar, biomass, and small hydro that are a part of DGs systems. Through private-sector investment, independent electricity producers (IPP) hope to add more megawatts to the power grid. The allocation for coal project procurement from IPPs is around 2500 MW [4]. While the Grand Inga Project, which aims to secure 2500 MW, is still under construction, South Africa and the DRC have already inked an energy deal. A legal basis for collaboration between the two nations is provided by the 2014 agreement.

The implementation of open-energy markets in many countries in the early 1990s paved the way for new competitors to enter the market, while the introduction of new electricity production led to its liberalization. In terms of the environment, several traditional types of generators release carbon dioxide, which could contribute to global warming. Changing from fossil fuels such as coal, gas, and oil to renewable energy sources such as solar and wind would reduce emissions. Governments have introduced incentives to encourage IPP to use renewable energy sources as an alternative source of energy [5].

Since electricity generated is to be consumed immediately as it cannot be efficiently stored, power system operators must make sure to balance power generation and demand. However, the redial design of distribution systems with the integration of DGs was not originally considered in the course of power system design [5]. This makes the involvement of the DGs essential to meet system technical and economical demands by optimally placing and sizing DGs. Optimal allocation and sizing find their main application in this regard in the use of different algorithms for decision-making.

### 1.2. Problem Statement

Optimization methods for electrical distribution network reconfiguration and DG placement have shown remarkable results for power loss minimization and voltage profile improvement. The optimization method used in this research paper for electrical distribution network reconfiguration is the HGAIPSO, an artificial-intelligence-based approach that selectively locates the optimal location for a particle. The problem in this study consists of

determining the optimal size or rating of DGs to be injected into power systems at specific nodes and also to find the optimal allocation of DGs in power systems for reactive power control and power loss minimization. Details of the GA, PSO, and IPSO algorithms can be found in the literature. There are two categories of DG technologies: those based on fossil fuels and those based on renewable energy sources. Examples of DGs using fossil fuels are internal combustion engines, combustion turbines, and fuel cells. Examples of DGs that use renewable energy sources include wind turbines, solar, biomass, geothermal, small hydro, and others. It is critical to evaluate the technological effects of DG in power networks. It takes time to examine the technical effects of DG on power networks [3]. As a result, DGs installed in electrical distribution networks need to be linked in a way that prevents power losses and changes in voltage profile. In this paper,

- The candidate bus in power systems at which DGs should be allocated based on the load flow, reactive power control limit, and power loss sensitivity factors is determined.
- The load flow problem, including constraints, is solved for DGs placement and sizing using a hybrid approach that combines both GA and IPSO. A sample electrical distribution network, i.e., the IEEE-30-bus test system, is used to evaluate the performance of the proposed algorithm.

*1.3. Research Aims and Objectives*

This research paper presents an improved optimization method for electrical power network reconfiguration applications. The iteration process of the algorithm takes account of constraints such as the real power loss minimization, reactive power limit, and voltage amplitude stability. Other aims of this study include the following:

- Development of the HGAIPSO algorithm, which is a combination of GA and IPSO algorithms, for optimal allocation of DGs in electrical distribution network. Formulation of a multi-objective function that incorporates the real power loss reduction index (PLRI), the reactive power control reduction index (QCRI), and the voltage profile improvement index (VPII) is provided.
- Reduction of power losses and improvement of voltage profiles by optimal sizing and placement of DGs.
- Comparative analysis of the proposed HGAIPSO algorithm with existing ones, i.e., the GA, PSO, and IPSO. An IEEE-30 bus electrical network with DGs allocation at various buses has been used for testing.

*1.4. Contributions of the Study*

This study will be advantageous to several parties, either directly or indirectly. Distribution firms and others will directly profit from the results of this study for the following reasons:

- Distribution firms and independent power producers (IPP) will be able to lessen actual power loss in their networks. By taking advantage of this decrease, they can avoid fines and reimbursements and increase their profit margins.
- Independent power producers (IPP) will be able to more simply and consistently incorporate small-scale renewable energy sources into their networks. This is crucial since the power generation sector is switching to green energy.
- Customers will gain, since they will feel comfortable using their devices with constant voltage profiles.

*1.5. Research Questions*

Answers to the following questions are provided via analysis of the sample electrical power network with DG allocation:

- By considering the real power losses and voltage profile as index for analysis, what would be the performance of a power system with and without optimal allocation and sizing of DGs?

- What are the advantages of using the HGAIPSO method compared to the GA, PSO, and IPSO methods in power system reconfiguration?

*1.6. Paper Structure*

The research paper is arranged into five sections. The introduction is the first section, which discusses generalities of electrical distribution networks. This section also discusses methods for minimizing power system losses and improving voltage profiles, as well as the manner in which DGs affect power system performance. Section 2 contains a literature review and discusses different optimization algorithms for power system analysis, presents a definition of sensitivity factors, defines the multi objective function equation, and includes a set of operational constraints. The reasons for choosing the proposed optimization algorithm are discussed in Section 3, the methodology section, which also provides details on the study of each different optimization strategy, including its parameters, implementation steps, flow charts, and the methods chosen for this investigation. Throughout Section 3, the proposed algorithm and its flowchart is described in detail. In Section 4, tables and graphs that present the results are provided with detailed discussion. A conclusion is given in Section 5.

## 2. Literature Review

### 2.1. Generality of Distributed Generation

Figure 1 shows the plot of a typical power loss versus DG size for a particular distribution network. As the DG size increases in power rating at a specific bus, the losses become minimal. If the size of DG increases further, it is likely to also see power losses increase. Therefore, to minimize the overall losses, DGs must be optimally allocated in a distribution network [6]. Figure 1 shows that it is not always appropriate to inject a large DG into an electrical distribution system. It should be of such size that it can consumed within the distribution substation boundary. A high-capacity DG in an electrical distribution network (DN) can result in reverse power flow and overvoltage at the point of common coupling (PCC) between DG and DN, resulting in very high loss [6,7]. Therefore, in choosing the rating of DG, it is important to take into account the size of the electrical distribution network, including the size of the load (MW). At high DG capacities, losses are higher because the distribution system was initially designed to maximize power flow from the source end (substation) to the load and gradually decrease conductor sizes from the substation to the consumer point. Using high-capacity DG without a strengthening of the system will result in excessive flow of power through small conductors and consequently higher losses.

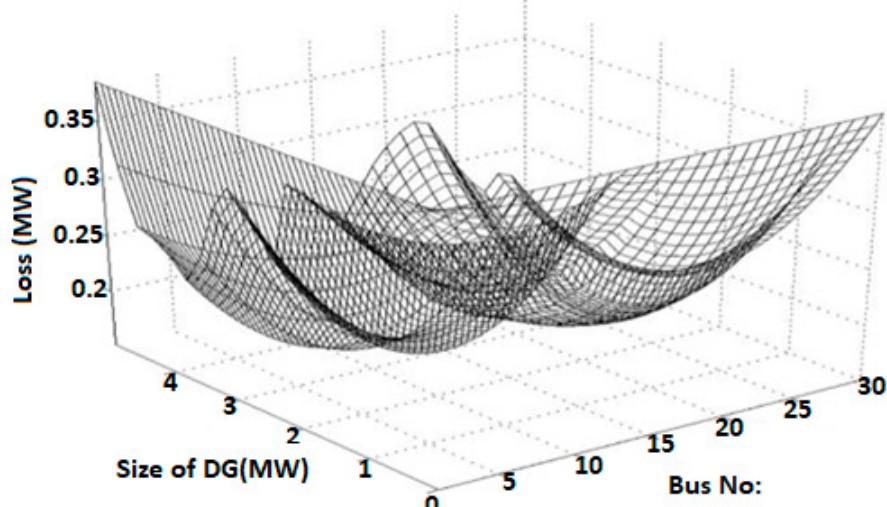

**Figure 1.** Effect of size and location of DG on system loss.

### 2.2. Radial Electrical Distribution Networks

Conventional electrical distribution networks are designed to operate in radial mode, as shown in Figure 2; the power is flowing from main generator to the consumer loads at low voltages. These types of electrical distribution networks can be easily reconfigured. The radial distribution system shown in Figure 2 has separated feeders that are coming from a single substation [8].

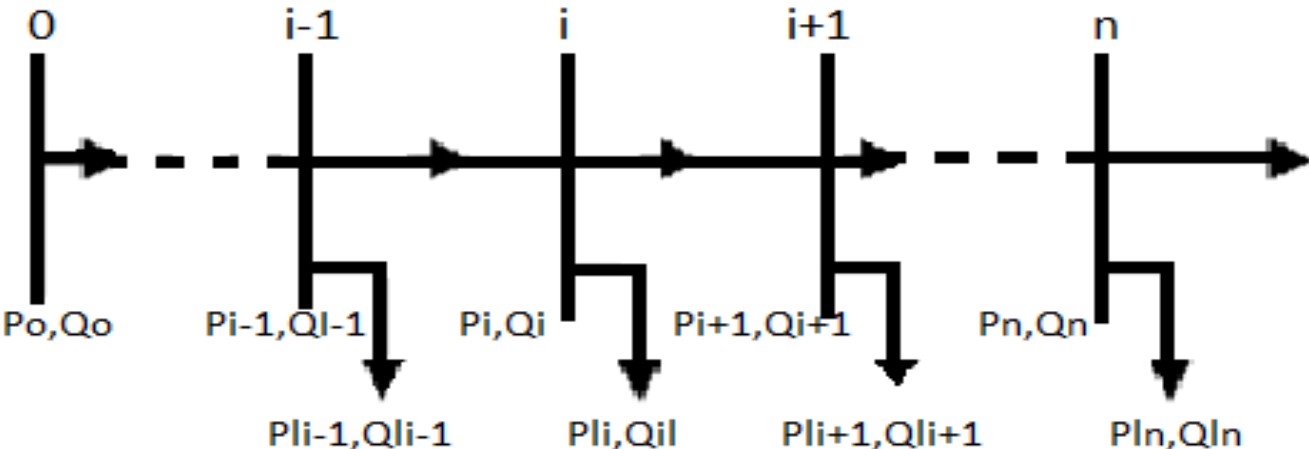

**Figure 2.** The line diagram for the radial distribution system.

### 2.3. Fuzzy Logic Algorithm

In [9], M. M. Aman and G. B. Jasmon proposed a new method for the placement of DG in radial distribution systems that employs fuzzy logic algorithms (FLA) in order to reduce real power losses and improve voltage profiles. FLA algorithms take a shorter time to compute than GA and PSO because they are simple by design. In [10], Nasim Ali Khan, S. Ghosh, and S. P. Ghoshal proposed an approach to allocate and size DGs in distribution networks and evaluated the benefits of employing DG by utilizing voltage profile improvement indexes (VPII) and line loss reduction indexes (LLRI).

For reactive power control with the aim of improving the voltage profile of power systems, the fuzzy logic algorithm shown in the Figure 3 flow diagram has been applied for many years. The voltage and controlling variables are converted into fuzzy sets to form the relations between voltage and controlling ability of the controlling device installed [11]. The control variables are selected based on local control towards a bus having unacceptable voltage and overall control towards the buses having poor voltage profile. They can be used in anything from small circuits to large mainframes. They can be used to increase the efficiency of the power distribution systems, as most of the data used in power distribution system analysis are approximate values and assumptions.

Advantages of Fuzzy Logic Algorithms

- Permanent and consistent.
- Easily documented.
- Easily transferred or reproduced.

Disadvantages of Fuzzy Logic Algorithms

- Unable to learn or adapt to new problems or situations.

### 2.4. Artificial Neural Network

These are biologically inspired systems that convert a certain set of inputs into a set of outputs by using a network of neurons in which each neuron can produce one output as a function of inputs. A fundamental neuron can be considered as a processor that can make a simple nonlinear operation of its inputs for the production of a single output. The understanding of the neuron operation and the pattern of their interconnection can be used

to form a computer for solving problems of classification of the pattern in the real world, as shown in Figure 4 [12]. They can be classified by their structures, such as the number of layers, topology, connectivity pattern, and recurrent.

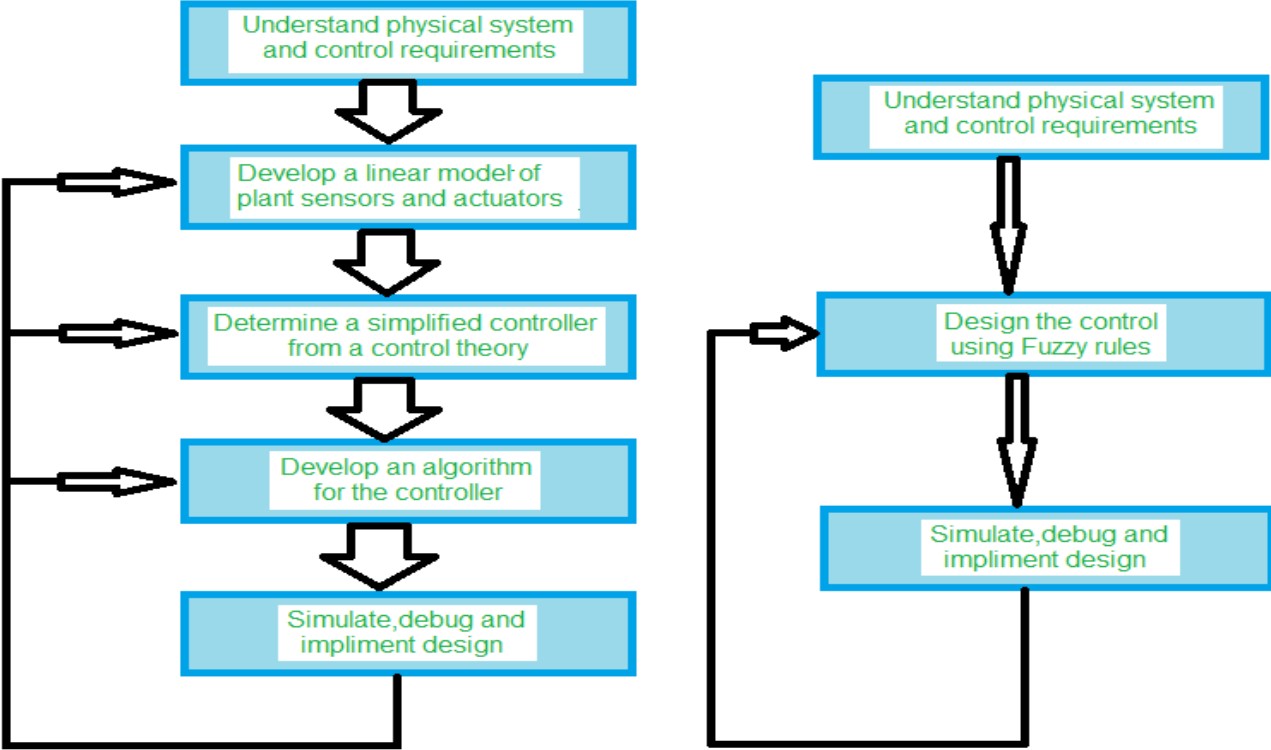

**Figure 3.** Fuzzy logic algorithm flow diagram.

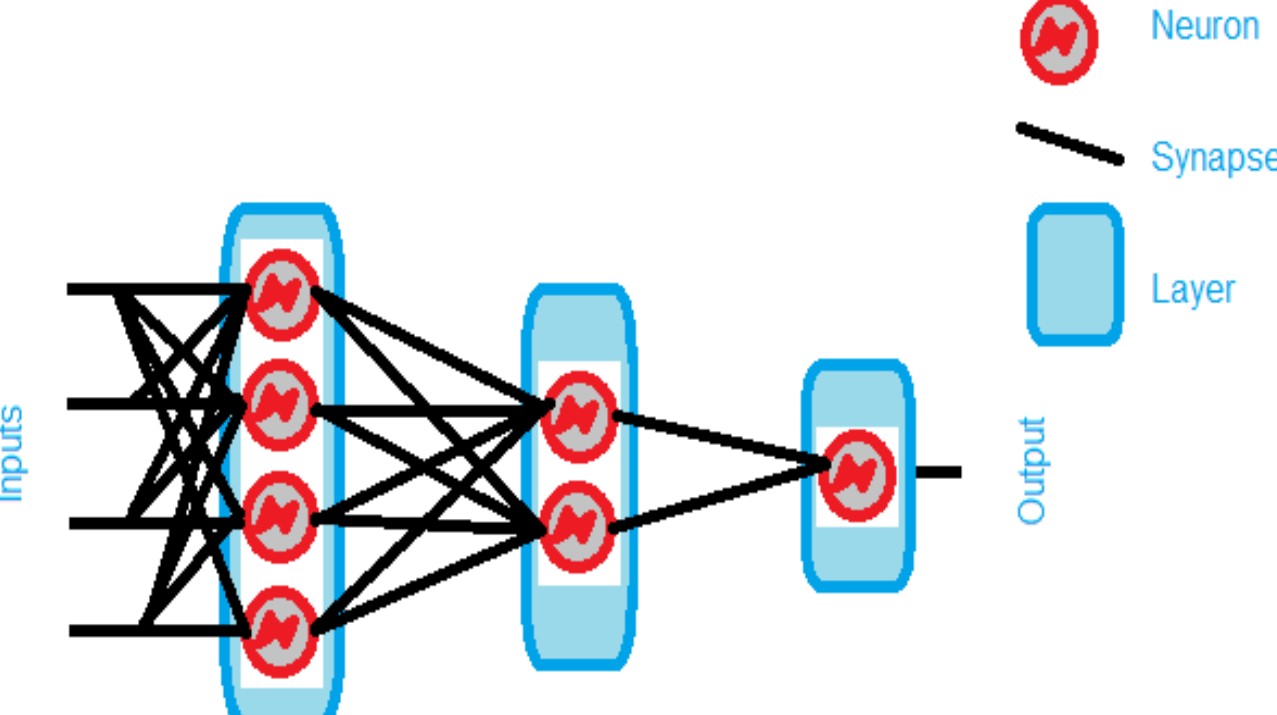

**Figure 4.** Architecture of a feed-forward ANN.

- Input Layer: The nodes are input units that cannot process the data and the information but can distribute this data and information to other units.
- Hidden Layers: The nodes are hidden units that are not visible. Their task is to provide the network ability to map or classify the nonlinear problems.
- Output Layer: The nodes are output units, which can encode possible values to be allocated to the case under consideration.

Advantages of Artificial Neutral Networks
- Speed of processing.
- They do not need specific details of the power system.

Disadvantages of Artificial Neutral Networks
- Large dimensionality
- Results can always be generated even if the input data are unreasonable.

In [13], the methodology proposed by M. Heydari, S.M. Hosseini, and S.A. Gholamian can solve the optimal capacitor allocation problem in practical distribution networks using an artificial neural network (ANN) algorithm. They proposed a best radial distribution network (BRDN) determined by an ANN algorithm. This algorithm finds the best location, size, and number of capacitor banks. In [12], M. Padma, N. Sinarami, and V.C Veera used the ANN algorithm to optimize the placement and sizing of DGs for capacity improvement and loss reduction.

### 2.5. Genetic Algorithm

In GA, the successive generations of a population adapt to the environment by mimicking the biological processes that occur in an ecosystem using the principle of evolutionary adaptation as modeled by genetic algorithms and unconstrained optimization methods. Natural selection and evolution is initiated by the genetic algorithm, which seeks to solve an optimization problem with objective function f(x) where x= x_1, x_2, x_3 . . . , x_n with optimization parameters in N-dimensions. The basic unit of the GA are genes and chromosomes, with standard optimization parameters encoded in binary code string. The GA genes represent binary codes and are combined together to make chromosomes [13].

In GA, the population represents candidate solutions as determined by n chromosomes; each chromosome represents a real value vector with m dimensions, where m is the number of optimized parameters. The flowchart of GA that is used to solve engineering problems is shown in Figure 5. The flowchart was constructed using the steps for the implementation of a genetic algorithm's steps.

In the methodology of [13], Auglt, R. Hooshmand, and M. Ataei determined the size and location of the DG units via genetic algorithms (GA). Cost-function-based methods give an optimal solution, but are computationally demanding and slow in convergence. While they have addressed the problem in terms of costs, cost-function calculations may lead to doubt regarding the exact size of DG units at suitable locations. In [14], Rahmat-Allah Hooshmand used a real-coded genetic algorithm (RCGA) to solve the problem of optimal placement of capacitor banks in unbalanced distributed systems with mesh/radial configurations. For reducing losses and controlling voltages of distribution systems, fixed and switched capacitors were optimally used.

In [15], S. Jalilzadeh, S. Galvani, H. Hosseinian, and F. Razavi developed a method based on RCGA (real-coded genetic algorithms) for finding the optimal values for fixed and switched capacitors in distribution networks. Various types of capacitors available on the market were used to model the loads at different load levels to solve for the optimal capacitor rate. In [16], Mehrdad Movahed examined voltage profiles in distribution systems and used reactive power injection to facilitate the improvement of voltage profiles in end busses that were far from slack buses. To determine optimum reactive power injection values, a genetic algorithm was used. The method resulted in an improved voltage profile and a decrease in losses.

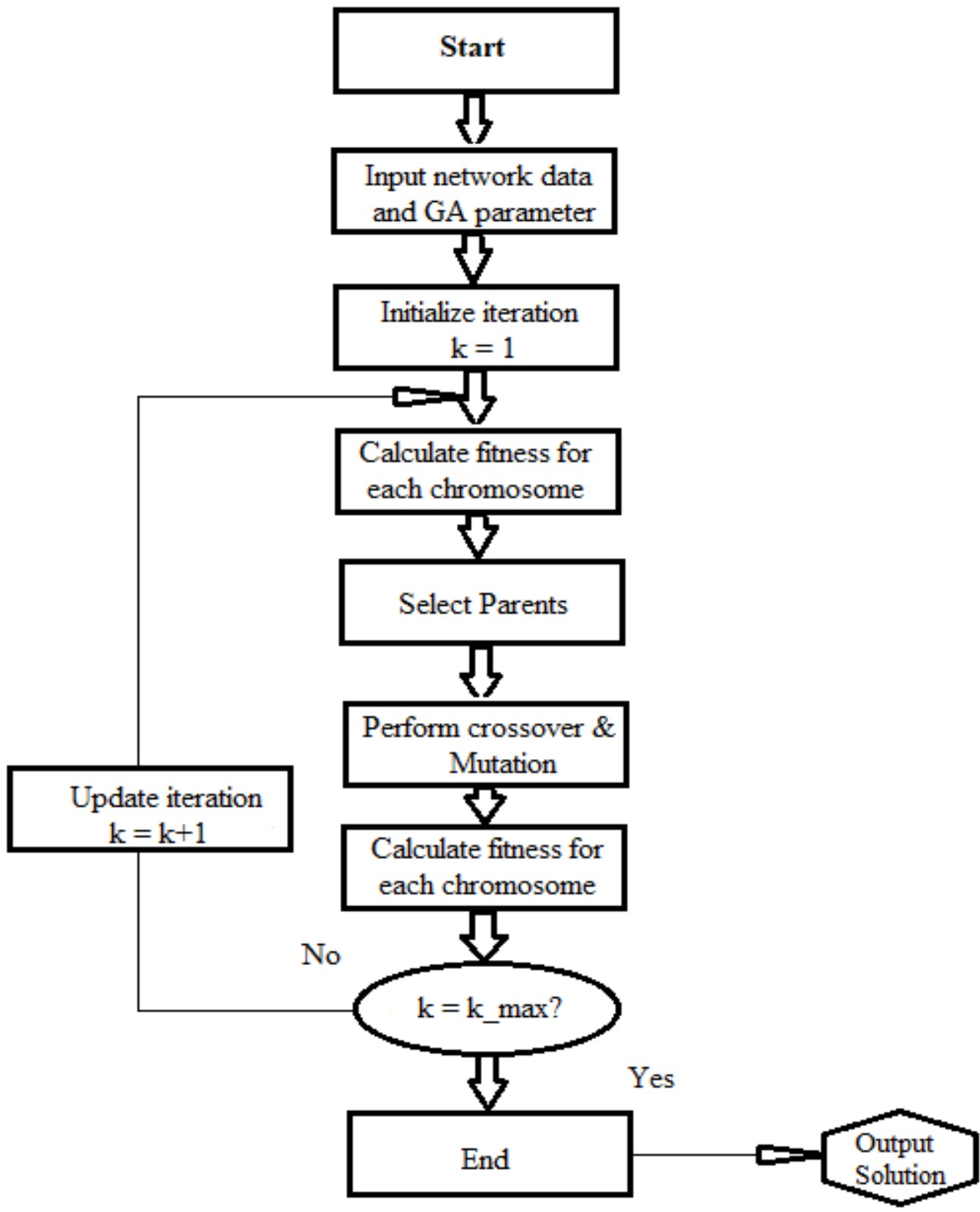

**Figure 5.** Steps for the implementation of a genetic algorithm.

In [17], according to Carpinelli, DG can be optimally located in radial distribution networks if they are located with the lowest minimum system losses. A GA solution was obtained by formulating the problem as an optimization problem with the objective of minimizing real power loss under equality and inequality constraints. Location was

determined based on how sensitive active power loss was to real power injection through DG. They demonstrated that the benefit increases as more locations are located within certain areas, beyond which it is not economically feasible. Only active power loss was considered in this formula.

In [18], a study by Saeed Amin Hajizadeh and Ehsan Hajizadeh Poor on shunt capacitor placement and distributed generation plants, reported the effect of genetic algorithms on the reduction of power loss and voltage profile in radial distribution systems. In the authors' research, they found that locating distributed generation plants and capacitors at the best locations leads to voltage profiles with lower losses. A distribution plant located near the load is the best location for shunt capacitors.

*2.6. PSO Algorthm*

PSO is an optimization method inspired by social behavior of birds flocking and fish schools as a starting point; as shown in the steps depicted in Figure 6, the PSO algorithm creates a population of particles that are positioned randomly throughout the search space. Particles represent solutions to the problem and have fitness values, with optimization being based on their fitness. Eventually, particles will move toward the optimal position, as they will have experienced their best position and the best solution. The updated velocity of particles is based on three factors: their past velocity, their best position to date, and the best position the entire swarm has reached in the past.

In [19], it is suggested by Amin Hajizadeh and Ehsan Hajizadeh that distribution planning can be done in a PSO-based manner. They developed a multi-objective setting for the optimal sizing and placing of distributed generation assets in distribution systems, minimizing the costs of power loss. Based on a PSO and weight approach, a best compromise was achieved between these two costs using the implemented technique. In [20], a study by Kai Zou and A. P. Agalgaonkar on shunt capacitors and DG units sought to establish voltage support zones in distribution networks. Their method reduced the search space by identifying the target voltage zones numerically and analytically. The authors proposed minimizing the investment cost for DG units and shunt capacitors by strategically placing DG units and shunt capacitors using PSO for overall voltage support and power loss reduction.

In [21], a methodology was proposed by I. Ziari et al. for the optimal allocation of capacitors and sizing capacitors for minimizing transmission line losses and improving voltage profiles. The results showed that the proposed methodology was more accurate and robust compared to genetic algorithm and nonlinear programming. In [22], according to Khanjanzadeh et al., the role of the location and capacity of a DG on enhancing stable voltage in radial distributed systems via PSO could be determined, and results of the PSO algorithm compared to the GA algorithm in terms of accuracy and convergence were discussed. PSO was found to be more accurate and faster to convergence than the GA method.

In [23], using a PSO-based technique, Varesi proposed optimization of DG unit allocation in the power system to reduce power losses and improve voltage profiles. Load flow algorithm and PSO were properly integrated to determine the best number, type, size, and placement of DG units. The researcher only considered two types of DG units. In [24], according to Mohammed M. and M. A. Nasab, a multi-objective PSO approach was used to optimize DG size and placement. The objective function used in the research was a hybrid objective function with two parts, the first part being a Power Loss Reduction Index, and the second part being a Reliability Improvement Index. Only acting power losses were considered in the research.

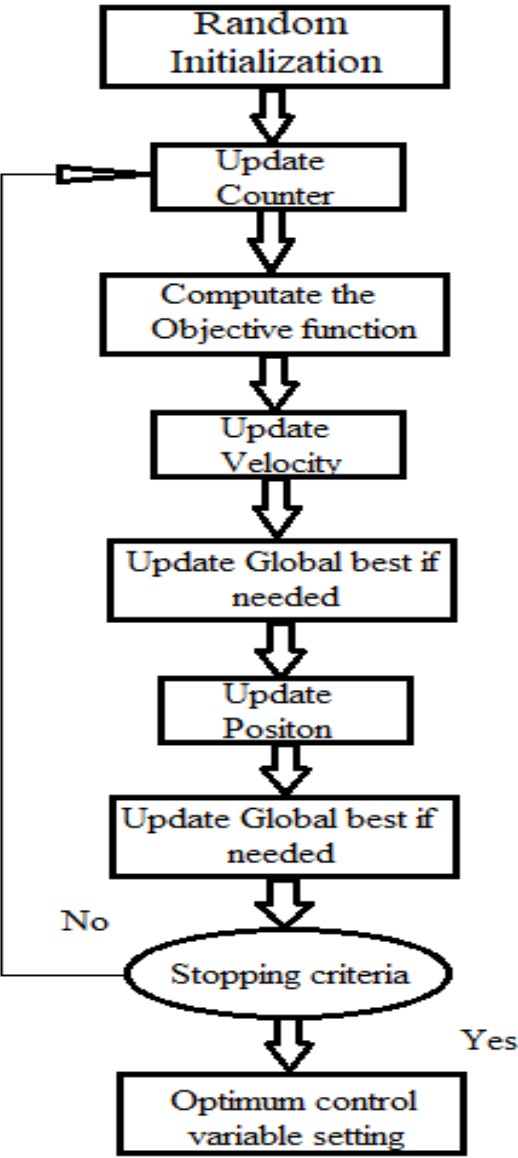

**Figure 6.** Flow chart of a particle swarm optimization algorithm.

In [25], using a Novel Binary Particle Swarm Optimization (NBPSO) technique, N. Mancer, B. Mahdad, and K. Srairi presented an improved total voltage profile by incorporating optimal placement of shunt capacitors with constraints in power distribution systems. The NBPSO method determined the optimal capacitor sizing and locations by using the near-global optimization approach. Shunt capacitors were incorporated into the sizing and placement of capacitors. In [26], a new study by Mehdi Nafar used discrete particle swarm optimization (DPSO) to optimize distribution system voltage profiles and to reduce total harmonic distortion (THD) in a distributed generation and capacitor system. The objective function had a term which prevented harmonic resonance between capacitor reactance and system reactance. Constraints included voltage limit, voltage THD, and number/size of capacitors and generators. The IEEE 33-bus test system was modified and employed in testing the proposed algorithm.

### 2.7. IPSO Algorrthm

In [27], the IPSO (IPSO) referred to by I. Ziari and G. Platt used optimal scheduling of DG and capacitor banks to minimize the reliability and line loss costs as well as the investment cost associated with electricity networks. They used crossover and mutation

operators in their research to reduce the likelihood of catching in the local smallest amount. They only considered true power losses when modeling IPSOs by minimizing power loss and maintaining the voltage profile and stability margin. In [28] N. Jain, S.N. Singh, and S.C. Srivastava developed a method for optimizing the placing and sizing of multiple DGs using IPSO. The researchers found that the method performed better compared to other classical and analytical methods for the placement of a single DG.

In [29], the IPSO based method proposed by Umapathi Reddy et al. is applied to loss reduction in unbalanced radial distribution systems. The study presented an efficient algorithm for determining where, what kind, and what size capacitor bank to install in unbalanced radial distribution systems. In addition, a selected bus identification method described for determining optimal capacitor placement locations using power loss indices (PLI) analysis. In unbalanced radial distribution systems, we used the IPSO approach to determine the optimal capacitor bank sizing. There are n particles in the population of the IPSO algorithm, and each represents a candidate solution; m is the number of optimized parameters for each particle, and each particle is an m dimensional real value vector. These parameters represent dimensions of the problem space. There are several steps in the process of IPSO. The IPSO algorithm needs to be adapted to each type of optimization problem that it must solve.

### 3. Methodology

*3.1. IEEE-30 Bus Electrical Network*

The IEEE-30 bus test represents a portion of the American Electric Power System (in the Midwestern US). The model has these buses at either 132 or 33 kV. The IEEE-30 bus test does not have line limits. Figure 7 shows the line diagram of the test system, while Tables 1–3 show the bus load injection, reactive power limit, and line parameters of the IEEE-30 bus test system, respectively [30].

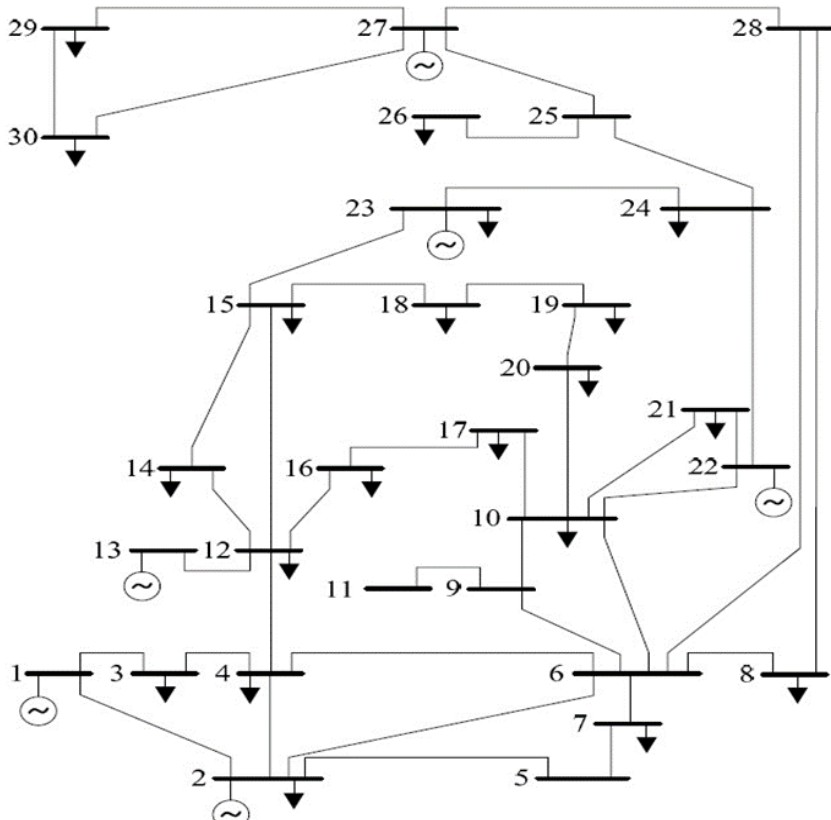

**Figure 7.** IEEE-30 bus test system.

**Table 1.** Bus load injection bus for IEEE-30 bus test system.

| Bus | Load (MW) | Bus | Load (MW) |
|---|---|---|---|
| 1 | 0.0 | 16 | 3.5 |
| 2 | 21.7 | 17 | 9.0 |
| 3 | 2.4 | 18 | 3.2 |
| 4 | 67.6 | 19 | 9.5 |
| 5 | 34.2 | 20 | 2.2 |
| 6 | 0.0 | 21 | 17.5 |
| 7 | 22.8 | 22 | 0.0 |
| 8 | 30.0 | 23 | 3.2 |
| 9 | 0.0 | 24 | 8.7 |
| 10 | 5.8 | 25 | 0.0 |
| 11 | 0.0 | 26 | 3.5 |
| 12 | 11.2 | 27 | 0.0 |
| 13 | 0.0 | 28 | 0.0 |
| 14 | 6.2 | 29 | 2.4 |
| 15 | 8.2 | 30 | 10.6 |

**Table 2.** Reactive power limit for IEEE-30 bus test system.

| Bus | Qmin (pu) | Qmax (pu) | Bus | Qmin (pu) | Qmax (pu) |
|---|---|---|---|---|---|
| 1 | −0.2 | 0.0 | 16 | | |
| 2 | −0.2 | 0.2 | 17 | −0.05 | 0.05 |
| 3 | | | 18 | 0.0 | 0.055 |
| 4 | | | 19 | | |
| 5 | −0.15 | 0.15 | 20 | | |
| 6 | | | 21 | | |
| 7 | | | 22 | | |
| 8 | −0.15 | 0.15 | 23 | −0.15 | 0.055 |
| 9 | | | 24 | | |
| 10 | | | 25 | | |
| 11 | −0.1 | 0.1 | 26 | | |
| 12 | | | 27 | −0.055 | 0.055 |
| 13 | −0.15 | 0.15 | 28 | | |
| 14 | | | 29 | | |
| 15 | | | 30 | | |

*3.2. Types of DGs and Number of DGs Used*

This study is about optimizing the location and size of three types of DGs using the assumptions that DGs are operating under any one of the three cases listed below [31].

- Case 1: DG that injects active power, which is only referred to as Type A (photovoltaic power); the number of DGs to be used will be determined by the proposed algorithm and one will be placed per selected bus.
- Case 2: DG that injects both active and reactive power, referred to as Type B (wind power); the number of DGs to be used will be determined by the proposed algorithm and one will be placed per selected bus.
- Case 3: DG that injects active power and absorbs reactive power, referred to as Type C (hydro power); the number of DGs to be used will be determined by the proposed algorithm and one will be placed per selected bus.

**Table 3.** Line parameters for IEEE-30 bus test system.

| Line | From Bus | To Bus | R(p.u) | X(p.u) | Tap Ratio | Rating(p.u) |
|------|----------|--------|--------|--------|-----------|-------------|
| 1 | 1 | 2 | 0.00192 | 0.0575 | | 0.300 |
| 2 | 1 | 3 | 0.0452 | 0.1852 | 0.9610 | 0.300 |
| 3 | 2 | 4 | 0.0570 | 0.1737 | 0.9560 | 0.300 |
| 4 | 3 | 4 | 0.0132 | 0.4845 | | 0.300 |
| 5 | 2 | 5 | 0.0472 | 0.5215 | | 0.300 |
| 6 | 2 | 6 | 0.0581 | 0.4521 | | 0.300 |
| 7 | 4 | 6 | 0.0119 | 0.4152 | | 0.300 |
| 8 | 5 | 7 | 0.0460 | 0.5560 | | 0.300 |
| 9 | 6 | 7 | 0.0267 | 0.1737 | | 0.300 |
| 10 | 6 | 8 | 0.0120 | 0.0379 | | 0.300 |
| 11 | 6 | 9 | 0.0000 | 0.1983 | | 0.300 |
| 12 | 6 | 10 | 0.0000 | 0.1763 | | 0.300 |
| 13 | 9 | 11 | 0.0000 | 0.0414 | 0.9700 | 0.300 |
| 14 | 9 | 10 | 0.0000 | 0.1160 | 0.9650 | 0.650 |
| 15 | 4 | 12 | 0.0000 | 0.0820 | 0.9635 | 0.650 |
| 16 | 12 | 13 | 0.0000 | 0.0420 | | 0.320 |
| 17 | 12 | 14 | 0.1231 | 0.2080 | | 0.320 |
| 18 | 12 | 15 | 0.0662 | 0.2560 | | 0.320 |
| 19 | 12 | 16 | 0.0945 | 0.1304 | | 0.320 |
| 20 | 14 | 15 | 0.2210 | 0.1987 | | 0.160 |
| 21 | 16 | 17 | 0.0824 | 0.1997 | | 0.160 |
| 22 | 15 | 18 | 0.1070 | 0.1932 | | 0.160 |
| 23 | 17 | 19 | 0.0936 | 0.2185 | 0.9590 | 0.160 |
| 24 | 18 | 20 | 0.0324 | 0.1292 | | 0.320 |
| 25 | 19 | 20 | 0.0348 | 0.0680 | | 0.300 |
| 26 | 10 | 17 | 0.0727 | 0.2090 | 0.9850 | 0.300 |
| 27 | 10 | 21 | 0.0116 | 0.0749 | | 0.300 |
| 28 | 10 | 22 | 0.0116 | 0.1499 | | 0.160 |
| 29 | 10 | 22 | 0.1000 | 0.0236 | | 0.300 |
| 30 | 21 | 23 | 0.1150 | 0.2020 | | 0.160 |
| 31 | 15 | 24 | 0.1320 | 0.1790 | | 0.300 |
| 32 | 22 | 24 | 0.1885 | 0.2700 | | 0.300 |
| 33 | 23 | 25 | 0.2544 | 0.3292 | 0.9655 | 0.300 |
| 34 | 24 | 26 | 0.1093 | 0.3800 | | 0.300 |
| 35 | 25 | 27 | 0.0000 | 0.2087 | | 0.300 |
| 36 | 25 | 27 | 0.2198 | 0.3960 | | 0.300 |
| 37 | 28 | 29 | 0.3202 | 0.4153 | 0.9810 | 0.300 |

*3.3. Development of HGAIPSO Algorithm*

This proposed method is a hybrid of GA and Improved Particle Swarm Optimization IPSO for optimal DG allocation DG. The DG is located within the selected buses on the distribution system with the aim of reducing power system losses and improving voltage profile. The selected buses for DG location are selected based on power flow and power loss sensitivity factors. The proposed algorithm (HGAIPSO) is able to select quickly by reducing the number of iterations. The location of the DG is determined by HGAIPSO based on sensitivity factors [32].

IPSO receives some GA output containing DG locations and DG sizes for various solutions. This GA output is then used as the initial particle set by IPSO. This helps IPSO reach convergence faster. Optimal solutions are derived from genetic algorithms through IPSO. The following Figure 8 contains the implementation steps showing how DG units in the distribution system are optimally allocated using HGAIPSO.

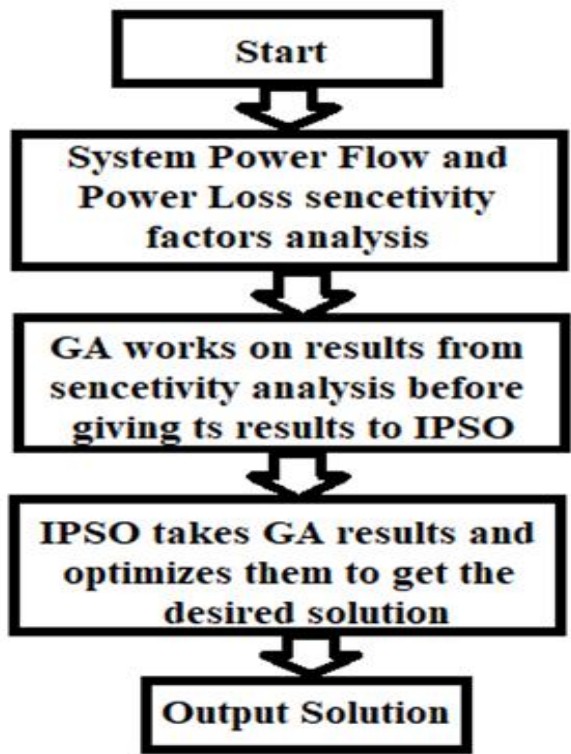

**Figure 8.** A summarized flow diagram for the proposed algorithm HGAIPSO.

*3.4. Formulation of System Power Flow Sensitivity Factors*

Specifically, the system flow sensitivity determines whether there is a change in the amount of power flowing in a transmission or distribution line between two buses, for example, bus $i$ and bus $j$, for the amount of power injected into any one of the buses in the system. Injection of complex power by a source into a system bus, such as some $i$th power system bus, can be described as follows [32]:

$$S_i = P_i + Q_i = V_i J_i^* - i = 1, 2, \ldots, n \tag{1}$$

where, with respect to ground, $V_i$ is the voltage at the $i$th bus.

$J_i^*$ Injected into the bus by the source current is given by

$$J_i = \sum_{j=1}^{n} Y_i V_{J;=1,2,\ldots n} \tag{2}$$

In other words, if we substitute this equation into the complex conjugate equation of power injection, we obtain the following:

$$P_i - jQ_i = \sum_{j=1}^{n} Y_i V_{J;=1,2,\ldots n} \tag{3}$$

We obtain the following equation when we combine the real and imaginary parts:

$$P_i = R_e \left\{ V_i * \sum_{j=1}^{n} Y_i V_J \right\} \tag{4}$$

$$Q_i = -\text{Im} \left\{ V_i * \sum_{j=1}^{n} Y_i V_J \right\} \tag{5}$$

The polar forms $V_i$ and $Y_{ij}$ can be expressed as

$$V_i = |V_i| e^{j\delta i} \tag{6}$$

$$Y_i = |Y_i| e^{j\delta i} \tag{7}$$

The actual and reactive powers can be expressed in general by polar representations:

$$P_i = |V_i| \sum_{j=1}^{n} |Y_{ij}| \cos(\theta_{ij} + \delta_{ij}); i = 1, 2, \ldots, n \tag{8}$$

### 3.4.1. Change in Real Power Flow Analysis

Real power flow in a line $k$ connecting two buses $i$ and $j$ can be expressed as:

$$Q_i = -|V_i| \sum_{j=1}^{n} |Y_{ij}| \sin(\theta_{ij} + \delta_{ij}); i = 1, 2, \ldots, n \tag{9}$$

where $V_i$ and $V_j$ are the voltage magnitudes at buses $i$ and $j$, respectively, and $\delta_i$ and $\delta_j$ are the voltage angles at buses $I$ and $j$, respectively. Further, $Y_{ij}$ is the magnitude of the element of the $Y_{BUS}$ matrix, and $\theta_{ij}$ is the angle of the $ij$th element of the $Y_{BUS}$ matrix.

Mathematically, the real power flow sensitivity is written as

$$P_{ij} = V_i V_j Y_{ij} \cos(\theta_{ij} + \delta_{ij}) - V_i^* V_{ji} \cos(\theta_{ij}) \tag{10}$$

By neglecting higher order terms and using Taylor series approximation, the change can be expressed as

$$\begin{bmatrix} \frac{\Delta P_{ij}}{\Delta P_n} \\ \frac{\Delta P_{ij}}{\Delta Q_n} \end{bmatrix} \tag{11}$$

Based on the partial derivatives of real power flow with respect to variables $\partial$ and $V$, the coefficients in the above equation can be obtained as follows:

$$\Delta P_{ij} = \frac{\partial P_{ij}}{\partial \delta_i} \Delta Q_i + \frac{\partial P_{ij}}{\partial \delta_j} \Delta Q_j + \frac{\partial P_{ij}}{\partial \delta_i} \Delta V_i + \frac{\partial P_{ij}}{\partial \delta_j} \Delta V_i \tag{12}$$

$$\frac{\partial P_{ij}}{\partial \delta_i} = V_i V_j Y_{ij} \sin(\theta_{ij} + \delta_{ij}) \tag{13}$$

$$\frac{\partial P_{ij}}{\partial \delta_i} = -V_i V_j Y_{ij} \sin(\theta_{ij} + \delta_{ij}) \tag{14}$$

$$\frac{\partial P_{ij}}{\partial V_i} = V_j Y_{ij} \cos(\theta_{ij} + \delta_{ij}) - 2V_j Y_{ij} \cos(\theta_{ij}) \tag{15}$$

$$\frac{\partial P_{ij}}{\partial V_j} = V_j Y_{ij} \cos(\theta_{ij} + \delta_{ij}) \tag{16}$$

### 3.4.2. System Power Loss Sensitivity Factors

A real and reactive power controls sensitivity factor calculated from the circuit diagram is shown in Figure 9 below.

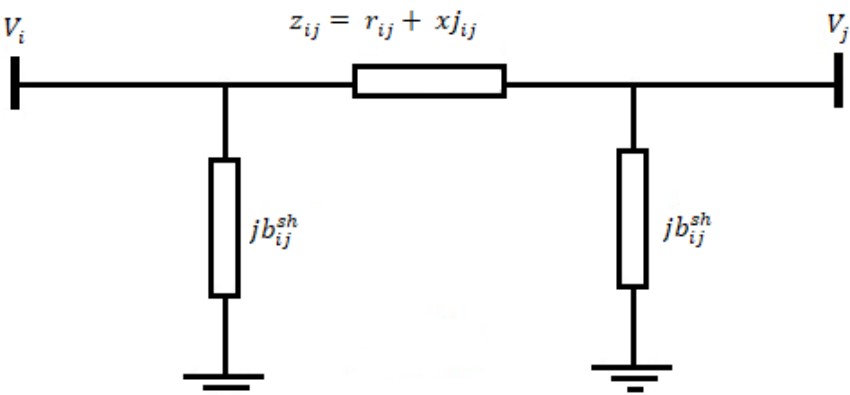

**Figure 9.** Circuit diagram of a line lumped model.

### 3.4.3. Choosing Weights Values for Multi-Objective Functions

The allocation of the various weights in a given multi-objective function vary according to the designer's concerns. This study gives more focus to real power loss reduction because it can result in a decrease in the total cost of operation and increase the efficiency of the power network [33]. However, as the other two factors are also important, a study of the effect of the weights on the fitness was conducted in order to determine the best weights combination to implement for the multi-objective function. In this study, weight values were assumed to be positive and were restricted as follows:

- $W_1$ was between 0.6 and 0.8
- $W_2$ and $W_3$ were restricted between 0.1 and 0.3

This was done to ensure that emphasis was given to the real power loss reduction index, as stated earlier, while still ensuring that all three indices are taken into consideration in the multi-objective function [34].

It is also important to note that the condition $|W_1| + |W_2| + |W_3| = 1$ has to be satisfied in each case. Weights are presented in Table 4.

**Table 4.** The effects of weights on fitness.

| Weight 1 (W1) | Weight 2 (W2) | Weight 3 (W3) | Best Fitness |
|---|---|---|---|
| 0.5 | 0.1 | 0.4 | 0.909 |
| 0.5 | 0.2 | 0.3 | 0.910 |
| 0.5 | 0.3 | 0.2 | 0.909 |
| 0.5 | 0.4 | 0.1 | 0.910 |
| 0.6 | 0.1 | 0.3 | 0.910 |
| 0.6 | 0.2 | 0.2 | 0.909 |
| 0.6 | 0.3 | 0.1 | 0.909 |
| 0.7 | 0.1 | 0.2 | 0.91 |
| 0.7 | 0.2 | 0.1 | 0.910 |
| 0.8 | 0.1 | 0.1 | 0.909 |

Table 4 shows the combinations of weights chosen, which are those that gave the minimum best fitness. The multi-objective function was given by

$$MOF = 0.6 PLRI + 0.2 QLRI + 0.2 \, VPII \tag{17}$$

The Newton Raphson method was used to estimate the base case reactive power, controlled to be 68.8881 MVAR, in order to ensure fair comparison. The number of DGs to be located and sized optimally was equal to the number of DGs in the comparison work; that is,

- the real power limitation for Type A, B, and C DGs is 0–12 MW;
- for Type B and C DGs, the reactive power limitation is 0–3 MVAR;
- for Type C, the reactive power limitation is −3–0 MVAR.

## 4. Results, Analysis, and Discussion

Based on the combined sensitivity factors of all the buses, the buses with a combined sensitivity factor greater than 0.80 were taken as the selected buses in order to determine the optimal location, size, and number of the DGs. The combined sensitivity factors of all the buses analyzed and the results are shown in Table 5.

**Table 5.** Results for CSF, fitness, and optimal DG sizes for selected buses.

| Selected Bus | Combined Sensitivity Factor (CSF) | DG | |
|---|---|---|---|
| | | Fitness | DG Size (MW) |
| 10 | 0.878 | 0.917 | 11.98 |
| 11 | 0.923 | 0.919 | 11.981 |
| 15 | 0.835 | 0.916 | 11.505 |
| 17 | 0.873 | 0.917 | 11.505 |
| 18 | 1.02 | 0.913 | 11.998 |
| 19 | 1.095 | 0.911 | 11.709 |
| 20 | 1.063 | 0.913 | 11.587 |
| 21 | 0.997 | 0.912 | 11.339 |
| 22 | 1.055 | 0.916 | 11.987 |
| 23 | 0.990 | 0.912 | 11.710 |
| 24 | 1.034 | 0.912 | 11.995 |
| 25 | 0.874 | 0.917 | 11.523 |
| 26 | 1.006 | 0.922 | 11.824 |
| 30 | 0.811 | 0.909 | 11.706 |

*4.1. Case 1, Type A DG*

Based on the columns representing fitness and DG size in Table 6, four optimal locations for the DGs of type A and their corresponding optimal sizes were selected. These locations gave the minimum fitness values and the corresponding DG sizes. According to their effectiveness, the four best locations and their optimal DG sizes are as follows:

- Bus number 19, at 11.7099 MW
- Bus number 21, at 11.9937 MW
- Bus number 24, at 11.9960 MW
- Bus number 30, at 11.7061 MW

**Table 6.** A comparison of results obtained using Type A DG.

| Method | Bus Number | DG Size | Power Losses | | Power Loss Reduction | | %Power Loss Reduction | |
|---|---|---|---|---|---|---|---|---|
| | | MW | MW | MVar | MW | MVar | %MW | %MVar |
| Without DG | | | 17.8798 | | | | | |
| GA | 10 | 11.472 | 13.3919 | - | 4.4879 | - | 25.1002 | - |
| | 10 | 11.904 | | | | | | |
| | 19 | 11.052 | | | | | | |
| | 24 | 11.772 | | | | | | |
| PSO | 10 | 11.694 | 12.2622 | - | 5.6176 | - | 31.4187 | - |
| | 15 | 11.394 | | | | | | |
| | 20 | 11.378 | | | | | | |
| | 30 | 10.577 | | | | | | |
| IPSO | 10 | 11.625 | 12.1851 | - | 5.6947 | - | 31.8499 | - |
| | 10 | 11.956 | | | | | | |
| | 22 | 11.995 | | | | | | |
| | 30 | 11.986 | | | | | | |
| HGAIPSO | 19 | 11.7099 | 10.6020 | - | 6.2778 | - | 40.7040 | - |
| | 21 | 11.9937 | | | | | | |
| | 24 | 11.9960 | | | | | | |
| | 30 | 11.7061 | | | | | | |

In order to determine the associated power losses and voltage levels, Newton Raphson method was used with the chosen four DG sizes and locations. The results obtained for power losses were compared to those obtained by other methods.

From Table 6 and Figure 10 for type A DG, it is clear that the HGAIPSO method reduced real power loss the most when compared with all other methods: GA by 25.1002%,

PSO by 31.4187%, IPSO by 31.8499%, and the proposed HGAIPSO method by 40.7040%. DG obtained from the proposed method demonstrated good results with DG allocations for loss reduction compared to the results obtained from other techniques. The HGAIPSO method is superior to the GA, PSO, and IPSO methods when determining the optimum location and size for a type A DG with the objective of reducing power losses within the electrical distribution system.

Figure 11 compares voltages for the case without DGs and with DGs optimally located and sized based on GA, PSO, IPSO, and HGAIPSO. A DG can affect the voltage stability of an IEEE-30 bus system, even though the voltages in an IEEE-30 bus system are within the acceptable limits, namely 0.95 pu to 1.1 pu. This can be seen in Figure 11: the inclusion of DGs does not result in voltage levels to be outside of acceptable limits. It is evident that all the bus voltages were within a range of 0.95 pu to 1.1 pu. The HGAIPSO method improved the voltage levels of the bus that had voltages below 1.0 pu to at least 1.01 pu, and no voltage exceeded the acceptable limit.

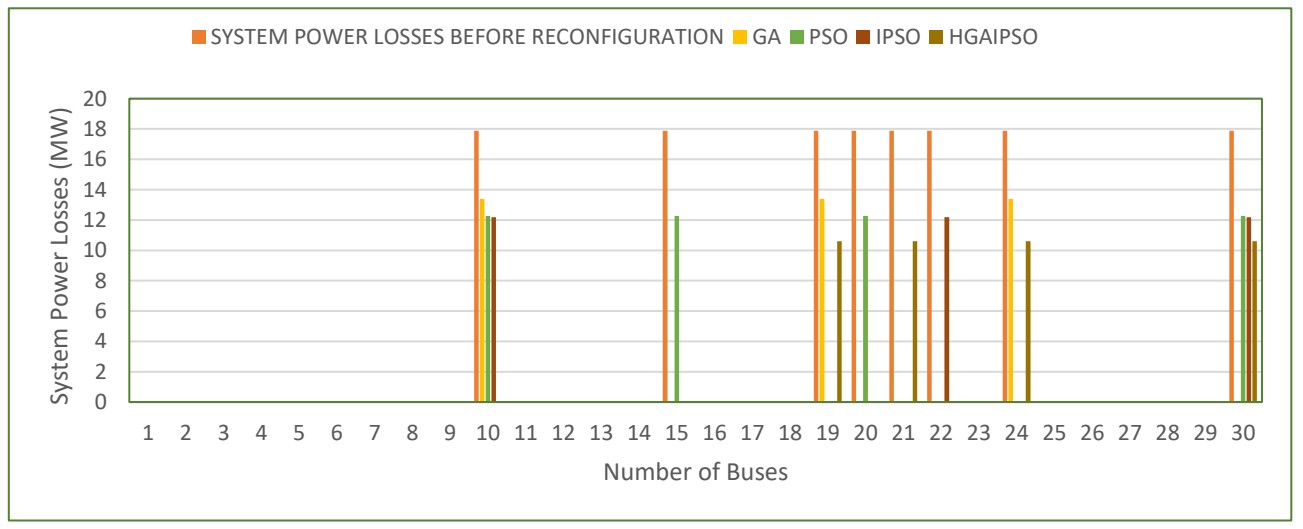

**Figure 10.** Comparison of results for power loss obtained using Type A DG.

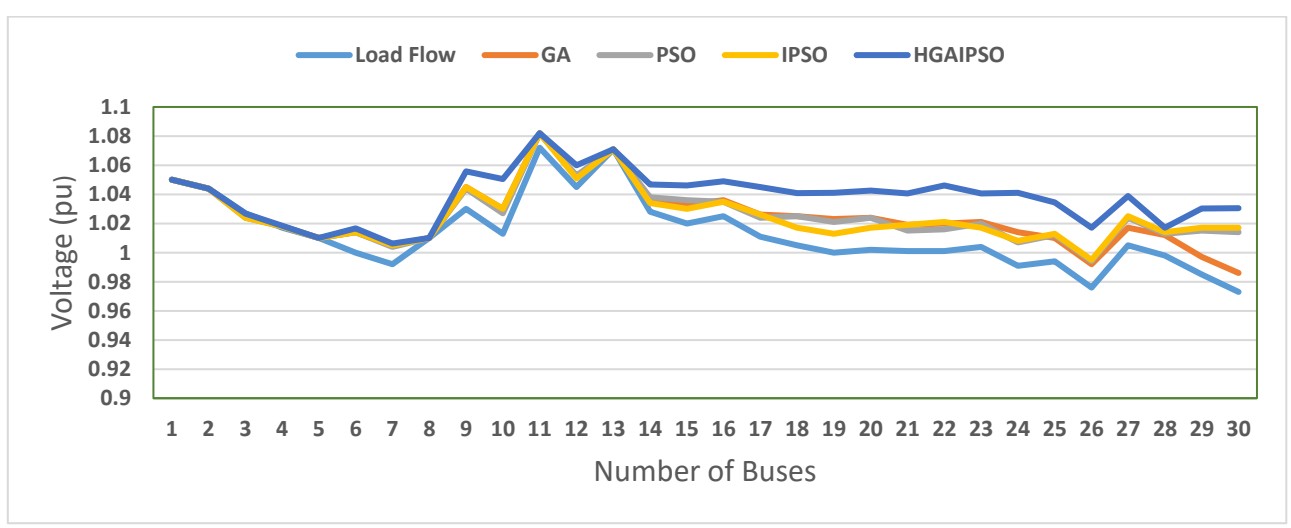

**Figure 11.** Bus voltage results for profile comparison using Type A DG.

*4.2. Case 2, Type B DG*

Based on the columns representing fitness and DG size in Table 7, four optimal locations for the DGs of type B and their corresponding optimal sizes were selected. These

locations gave the minimum fitness values and the corresponding DG sizes. According to their effectiveness, the four best locations and their optimal DG sizes are as follows:

- Bus number 19 with a DG generating 11.7872 MW and 2.9609 MVar
- Bus number 23 with a DG generating 11.7548 MW and 3.0002 MVar
- Bus number 24 with a DG generating 12.0001 MW and 1.3702 MVar

According to Table 7 and Figure 12, using the HGAIPSO method to optimize the location and size of this type of DG results in a 36.2403% reduction in real power losses. This is higher in comparison to GA, PSO and IPSO with reductions of 31.5890%, 32.2923%, and 33.1648%, respectively. This is also comparable to the sizes determined using other techniques that we chose for the DGs sizing and allocation for power loss minimization.

**Table 7.** Comparison of bus voltage using Type B DG.

| Method | Bus Number | DG Size | Power Losses | | Power Loss Reduction | | %Power Loss Reduction | |
|---|---|---|---|---|---|---|---|---|
| | | MW | MW | MVar | MW | MVar | %MW | %MVar |
| Without DG | | | 17.8798 | | | | | |
| GA | 10 | 11.35 + j1.22 | 12.2260 | - | 5.6538 | - | 31.5890 | - |
| | 23 | 11.47 + j1.17 | | | | | | |
| | 24 | 11.92 + j2.04 | | | | | | |
| | 30 | 11.816 + j1.468 | | | | | | |
| PSO | 10 | 11.474 + j2.159 | 12.1060 | - | 5.7738 | - | 32.2923 | - |
| | 17 | 11.981 + j0.919 | | | | | | |
| | 20 | 11.67 + j2.309 | | | | | | |
| | 30 | 11.349 + j3 | | | | | | |
| IPSO | 10 | 11.83 + j0.001 | 11.9500 | - | 5.9298 | - | 33.1648 | - |
| | 21 | 11.433 + j3 | | | | | | |
| | 24 | 11.739 + j3 | | | | | | |
| | 30 | 11.995 + j0.001 | | | | | | |
| HGAIPSO | 19 | 11.7872 + j2.9609 | 11.4001 | - | 6.4797 | - | 36.2403 | 24.2585 |
| | 23 | 11.7548 + j3.0002 | | | | | | |
| | 24 | 12 + j1.3702 | | | | | | |
| | 30 | 11.8308 + j1.5817 | | | | | | |

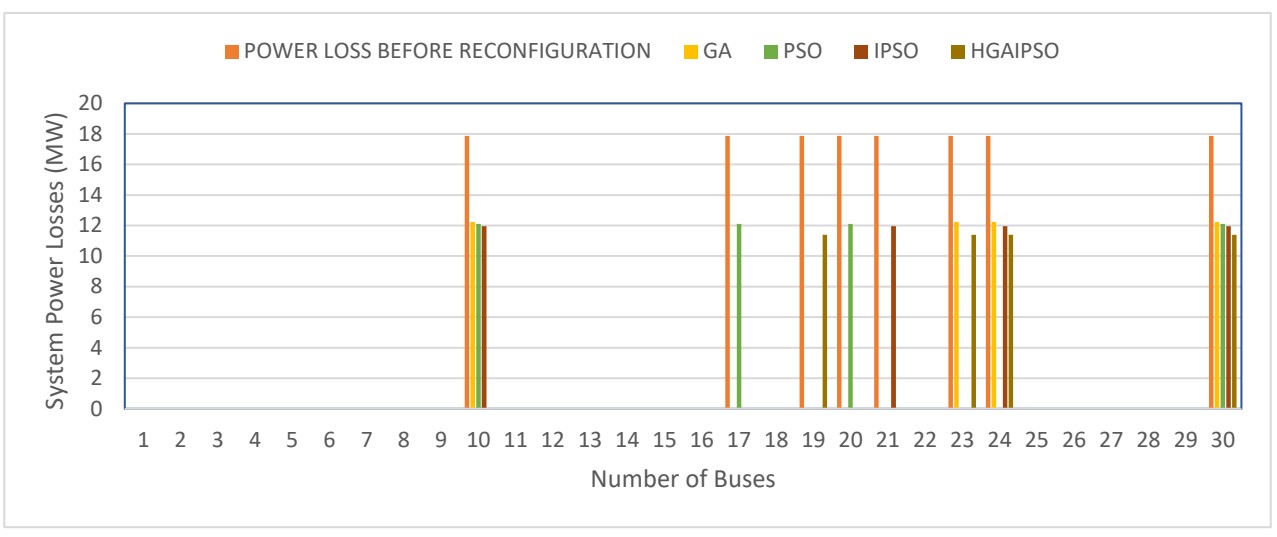

**Figure 12.** A comparison of results for power loss obtained using Type B DG.

Voltage Profile

An analysis of the voltage profile of the IEEE-30 bus system was performed after the placement and sizing of the type B DGs were optimized. Figure 13 below shows the results of the bus voltage levels under this condition. A comparison is also given between this case and the one without DGs and with DGs placed and sized using other methods.

Figure 13 compares voltages for the case without DGs and with DGs optimally located and sized based on GA, PSO, IPSO, and HGAIPSO. A DG can affect the voltage stability

of an IEEE-30 bus system, even though the voltages in an IEEE-30 bus system are within the acceptable limits, namely 0.95 pu to 1.1 pu. The Figure 13 shows that the inclusion of DGs did not result in voltage levels outside of acceptable limits. It is evident that all the bus voltages were within a range of 0.95 pu to 1.1 pu. HGAIPSO method improved the voltage levels of the bus that had voltages and no voltage exceeded the acceptable limit.

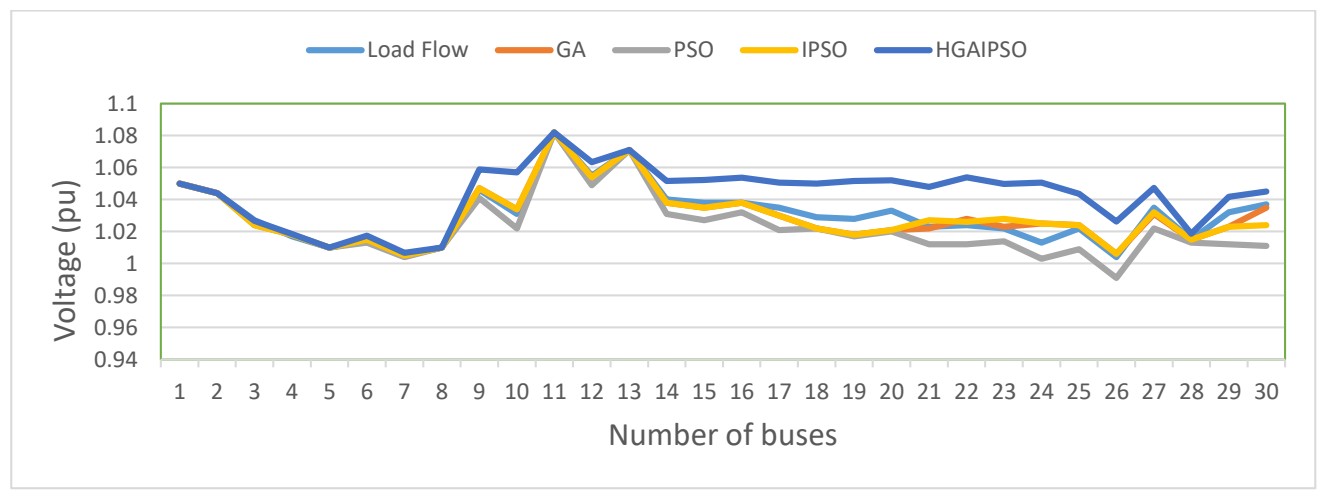

**Figure 13.** Bus voltage profile comparison using Type B DG.

### 4.3. Case 3, Type C DG

Based on the columns representing fitness and DG size in Table 8, four optimal locations for the DGs of type B and their corresponding optimal sizes were selected. These locations gave the minimum fitness values and the corresponding DG sizes. According to their effectiveness, the four best locations and their optimal DG sizes are as follows:

- Bus number 19, with a DG generating 12.0010 MW and absorbing 0.4882 MVar
- Bus number 24, with a DG generating 11.9470 MW and absorbing 0.5042 MVar
- Bus number 21, with a DG generating 11.9179 MW and absorbing 0.0692 MVar

**Table 8.** A comparison of results obtained using Type C DG.

| Method | Bus Number | DG Size | Power Losses | | Power Loss Reduction | | %Power Loss Reduction | |
|---|---|---|---|---|---|---|---|---|
| | | MW | MW | MVar | MW | MVar | %MW | %MVar |
| Without DG | | | 17.8798 | | | | | |
| GA | 10 | $9.0384 - j0.0882$ | 11.5265 | - | 6.3533 | - | 35.6967 | - |
| | 18 | $11.1120 - j0.7150$ | | | | | | |
| | 22 | $11.7480 - j0.5891$ | | | | | | |
| | 30 | $10.0081 - j0.4870$ | | | | | | |
| PSO | 10 | $11.885 - j0.7970$ | 11.1056 | - | 6.7742 | - | 37.8874 | - |
| | 18 | $10.8811 - j0.3215$ | | | | | | |
| | 20 | $11.5631 - j0.8990$ | | | | | | |
| | 30 | $11.5310 - j0.3831$ | | | | | | |
| IPSO | 10 | $12.0215 - j0.5260$ | 11.2099 | - | 6.6699 | - | 37.3041 | - |
| | 19 | $10.8610 - j0.3002$ | | | | | | |
| | 22 | $11.9170 - j0.8370$ | | | | | | |
| | 30 | $11.9560 - j0.5260$ | | | | | | |
| HGAIPSO | 19 | $12.0010 - j0.4882$ | 10.2021 | - | 7.6777 | - | 42.9406 | 24.212 |
| | 21 | $11.9470 - j0.5042$ | | | | | | |
| | 24 | $11.9179 - j0.0692$ | | | | | | |
| | 30 | $11.3651 - j0.5807$ | | | | | | |

Table 8 and Figure 14 show the comparison of the results of the power losses as a function of the different methods. When compared to GA, PSO, and IPSO, the HGAIPSO method shows the greatest reduction in power loss at 42.9406%. The proposed method performed better than GA (35.6967%), PSO (37.8874%), and IPSO (37.3041%).

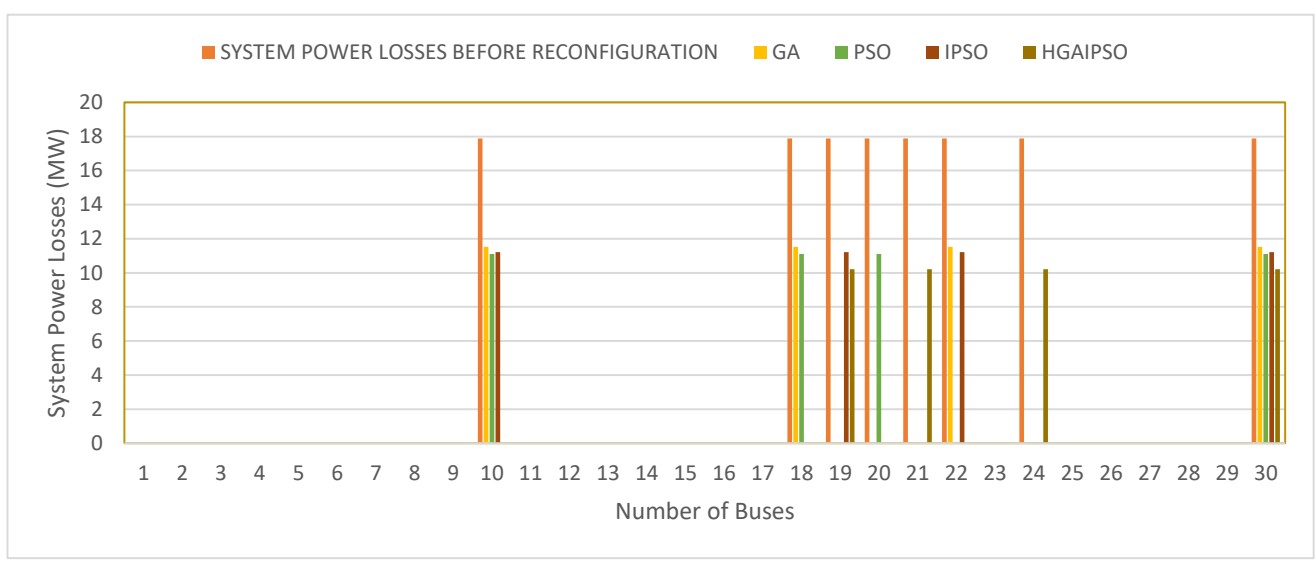

**Figure 14.** A comparison of results for power loss obtained using Type C DG.

The results from Figure 15 clearly show that the use of the HGAIPSO method resulted in significantly higher bus voltage, which means the inclusion of the DGs optimized their placement and sizing. Based on the optimization of type C DG location and size, it was possible to achieve a higher bus voltage level of 1.01 pu from 0.973 pu. This means that the highest value maintained at 1.095 pu. Therefore, based on these data, the bus voltage profile improved.

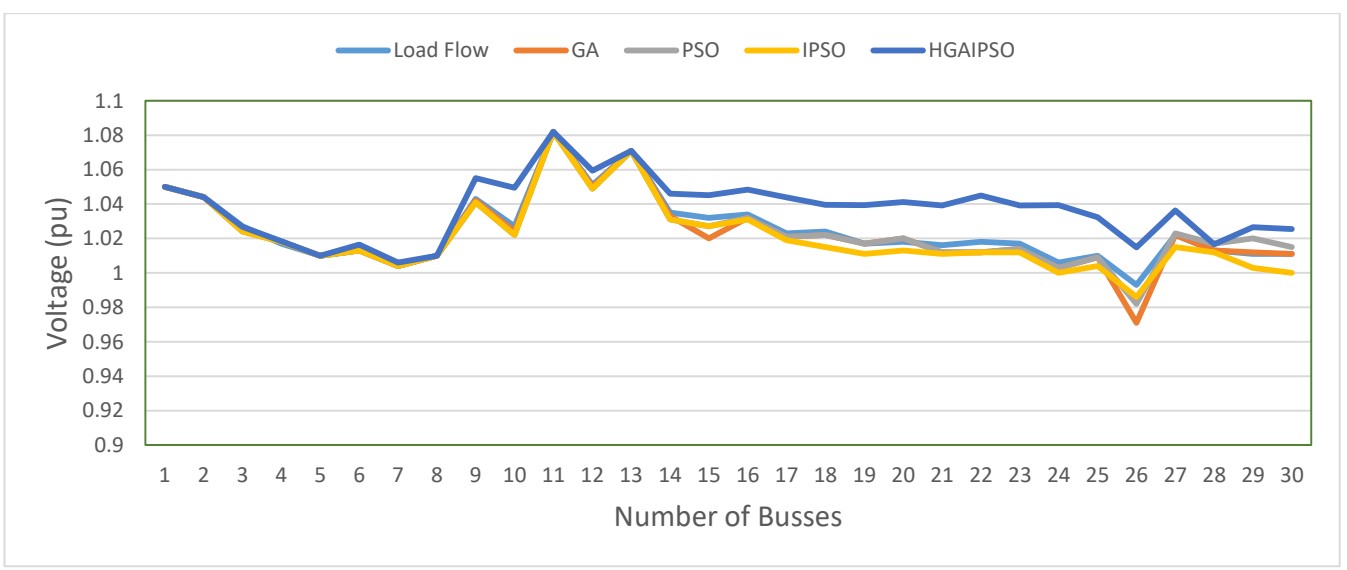

**Figure 15.** Bus voltage profile comparison using Type C DG.

## 5. Conclusions

By optimizing the location and size of DGs, the problem of power losses in systems was solved because power losses were reduced and voltage profiles were improved. This study presents a hybridized algorithm (HGAIPSO) to reduce system power losses and improve voltage profiles. Combining the sensitivity factors and test the on the IEEE-30 bus test system was effective in reducing the number of iterations for algorithms. For the IEEE-30 bus test system, 14 buses were selected as optimum DG locations. Comparing the HGAIPSO method to the GA, PSO, and IPSO method in three types of DGs using IEEE-30 bus showed that it can be reduced. Using Type, A, Type B, and Type C DGs, real power losses were reduced by 40.7040%, 36.2403%, and 42.9406%, respectively. Each of the three

cases produced the highest bus voltage of 1.01 pu, which shows that the voltage profile was generally improved.

The IEEE-30 bus test system's losses were decreased and its voltage profile may be improved as a consequence of HGAIPSO, proving that this approach is more suited to maximizing this parameter than GA, PSO, and IPSO. It was clearly shown through the use of the HGAIPSO algorithm how distribution generation affected power loss and voltage profile: there was a decrease in system power losses when distributed generators were added to the power system, up to an ideal number. Further DG introduction from the ideal number is also expected to cause the voltage profile to operate in a different way, which would impair bus voltages within the permissible range. The research objectives were met successfully and the HGAIPSO optimization algorithm implemented in this study proved to be more effective than GA, PSO, and IPSO for optimum locating and sizing of DGs in power distribution systems to minimize losses.

This study used a hybrid GA and PSO technique to solve the Distribution Network Reconfiguration problem in a more effective, accurate manner. This tactic makes use of a number of strategies to sustain population variety. Additionally, the system uses a mending approach to fulfill the radial specifications for each GA chromosome or PSO particle, greatly decreasing the solution space. The global optimum may be located using the suggested technique with a high rate of convergence and no premature convergence. The proposed hybrid strategy outperforms existing approaches in terms of computation time, average loss reduction, and least standard deviation while searching for optimum solutions of many independent runs.

*Recommendations for Future Work*

- Since the loading of the power network cannot be stopped, the power distribution companies need to apply this approach any time there is a need to incorporate DG in the power distribution network.
- Include other aspects of the power system, such as stability, and improve the multi-objective function.
- Code for this project programmed in Matlab resulted in long iteration times. Thus, further efforts will need to try to reduce these delays.
- Negative impacts of DG, which can be eliminated with optimal allocation of DG, should be investigated by planning engineers.

**Author Contributions:** Conceptualization, M.N. and K.M.; methodology, M.N.; software, M.N.; validation, M.N., K.M. and M.C.L.; formal analysis, M.C.L.; investigation, M.N.; resources, K.M.; data curation, M.N.; writing—original draft preparation, M.N.; writing—review and editing, M.C.L.; visualization, M.C.L.; supervision, K.M. and M.C.L.; project administration, K.M.; funding acquisition, K.M. All authors have read and agreed to the published version of the manuscript.

**Funding:** This research received no external funding and The APC was funded by Durban University of Technology.

**Informed Consent Statement:** Not applicable.

**Conflicts of Interest:** The authors declare no conflict of interest.

## Abbreviations

| | |
|---|---|
| CSF | Combined Sensitivity Factors |
| ANN | Artificial Neuron Network |
| DG | Distributed Generator |
| DGs | Distributed Generators |
| DISCO | Distribution Company |

| IPSO | Hybrid Particle Swarm Optimization |
|------|-------------------------------------|
| LLRI | Line Loss Reduction Index |
| HGAIPSO | Hybrid Genetic Algorithm Improved Particle Swarm Optimization |
| LSF | Loss Sensitivity Factor method |
| IPSO | Improved Particle Swarm Optimization |
| ORPF | Optimal Reactive Power Flow |
| PLI | Power Loss Index |
| PLRI | Real Power Loss Reduction Index |
| PSO | Particle Swarm Optimization |
| QLRI | Reactive power control Reduction Index |
| FLA | Fuzzy Logic Algorithm |
| GA | Genetic Algorithm |
| T&D | Transmission and Distribution |
| BRDN | Best Radial Distribution Network |
| DSTATCOM | Distribution-static compensator |
| RCGAA | Real-Coded Genetic Algorithm |
| NBPSO | Novel Binary Particle Swarm Optimization |
| DPSO | Discrete Particle Swarm Optimization |
| THD | Total Harmonic Distortion |
| PDIP | Primal Dual Interior Point |
| VSI | Voltage Sensitivity Indexes |

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
