# Peer review of "Power Loss Minimization and Voltage Profile Improvement by System Reconfiguration, DG Sizing, and Placement"

_computation, doi:10.3390/computation10100180_

Round 1
Reviewer 1 Report
The authors in the paper under review propose a reconfiguration methodology based on a hybrid optimization algorithm, consisting of a combination of the genetic algorithm and the improved particle swam optimization algorithm for minimizing active power loss and maintaining the voltage magnitude at about 1 p.u. The proposed reconfiguration methodology is tested in an IEEE-30 bus electrical network system with DGs allocations and the simulations are conducted using MATLAB software. The simulation results show that the proposed algorithm can be an efficient and promising optimization algorithm for distribution network reconfiguration problems. Overall the content of the paper deserves publication. My comments on this manuscript are:
1. The introduction should be more condensed, with more contemporary literature and applications.
2. What is the most important conclusion making the work publishable in this journal?
3. Please explain how your study advances or adds to the field's current level of knowledge and provide a convincing explanation for your effort.
Author Response
- The introduction has been proofread and some English errors has been corrected, the introduction has been added with some literature.
- Conclusion is refined and changed to give more details.
- The contribution of the study has been added on chapter one to address the reviewer’s comment.
Reviewer 2 Report
Improvement can be addressed to reference to other approaches not only on evolutionary algorithms like Bayesian network and introducing attribute relevance analysis to prove that traditional evolutionary algorithms have or have not alternative for such use cases.
Author Response
Changes in the introduction has been made to add more the background of the study and some references has been added to make the paper more publishable. In addition to the above comment, all spelling and grammatical errors pointed out by the reviewer have been corrected.